# Logic Channel Validation and Enhancement of Zero-Shot Vision-Language Comprehension on Vision Language Models

## Abstract

Frontier Large Vision-Language Models (LVLMs) exhibit remarkable capabilities in Visual-Language Comprehension (VLC) tasks, enabled by pretraining on vast visual-textual corpus. However, they are often deployed as zero-shot solution in a black-box manner, as retraining challenges remain due to data privacy or model inaccessibility. Validating and understanding the behavior of the models become important for generalization to new task. We propose a Logic Channel, in parallel with the black-box model channel, to perform explicit logic reasoning for validation and enhancement. The frontier LVLM, encapsulating latent vision-language knowledge, can be considered as an Implicit Logic Channel. The proposed Explicit Logic Channel, mimicking human logic reasoning, incorporates a Large Language Model (LLM), a Visual Foundation Model (VFM), and a logical reasoning module involving novel probabilistic inference for factual, counterfactual, relational, and causal condition reasoning over the extracted and grounded visual-textual facts. Cross-channel logic consistency analysis enables model validation and selection, even without ground-truth annotations. Additionally, cross-channel integration further improves performance in zero-shot tasks over SOTA models. Our experiments on three recent challenging VLC benchmarks, Neg-Bench, HC-RefCOCOg, and HC-RefLoCo, demonstrate the effectiveness of the proposed Logic Channel for logic-based model validation, selection and improvement on LVLM with enhanced explainability and trustworthiness.

## 1 Introduction

Large Language Models (LLMs) have demonstrated remarkable general intelligence in text-based tasks Zhao et al. (2025). Recently, LLMs have been extended as Multimodal LLMs (MLLMs) by incorporating visual inputs. LVLMs have advanced significantly, as evidenced by the frequent release of frontier models by leading AI organizations and the rapid growth of vision-language (VL) benchmarks in both data diversity and quantity Li et al. (2025a). While LVLMs have achieved advanced capabilities in visual language comprehension (VLC) across a wide range of multi-modal tasks, recent research increasingly highlight the limitations of LVLMs in terms of reliability, factuality, explainability, and logic reasoning Dang et al. (2024). Frontier LVLMs are often deployed for new tasks as black-boxes in zero-shot learning, due to data privacy concern, large model size, as well as inaccessibility of closed-source model. It is crucial to be able to identify reliable model and enhance the accuracy of predictions without re-training the model for new tasks Khan & Fu (2024).

Human make decision and judgment on facts and logical rules Wang et al. (2022). The faithfulness, reasonability, and trustworthiness of a decision are justified based on explicit and concrete facts, relations, and logic causation. On this observation, we propose a Logic Channel, in parallel with LVLM, to perform a logic validation of LVLM's prediction on logic consistency, as well as enhance the accuracy of LVLM's prediction. Since a LVLM has been pre-trained on vast vision-language datasets, it is assumed to have learned human logic on visual-language problems. It can be considered as an Implicit Logic Channel (ILC). However, human natural language is not a complete logic language Gomes (2024). The ground truth (gt) annotations based on natural language may be ambiguous, uncertain, or logically incomplete, leading to logically inconsistent predictions. To complement ILC, we propose an Explicit Logic Channel (ELC), where we prompt a LLM as Language

Parser to extract concept-level logic conditions from the input text, employ a VFM to ground the the facts explicitly in the image, and design a Logic Reasoning module with novel probabilistic inference approach to make decisions explicitly on the grounded facts, relations, and causal conditions. The consistency between the two logic channels can be exploited for model validation, justification, and selection, even when gt annotations are not available for evaluation. In addition, the combination of both channels further enhance the accuracy of prediction without additional fine-tuning.

To investigate the effectiveness of the proposed Logic Channel for VLC validation and accuracy enhancement under zero-shot setting, we perform experiments on three recent VLC challenges, i.e., NegBench, HC-RefCOCOg and HC-RefLoCo. Our experimental results show that, (a) the Logic Channel with probabilistic inference is able to produce accurate and robust predictions with explicit visual justification; (b) logic consistency between the ILC and ELC is effective for model validation and selection even without gt annotation; (c) the combination of ILC and ELC can further improve SOTA performance with enhanced explainability and trustworthiness.

The main contributions can be summarized as: (1) A Logic Channel on Foundation Models and Logic Reasoning for VLC tasks, producing concrete prediction on explicit facts and logic relations without training; (2) Logic consistency between ILC and ELC for model validation without the need of gt annotation; (3) Effective LC solutions with novel probabilistic inference approach for VLC with explicit logic reasoning on factual and counterfactual evidence, relations, and conditions of causation, achieving significant improvement over SOTA performance on three VLC benchmarks.

## 2 RELATED WORK

**LVLMs.** There is a growing number of LVLMs released in recent years Li et al. (2025b). One cluster are jointly embedding-based Vision Language Models (VLMs), where the visual and textual inputs are encoded separately, and then embedded into a shared latent space. Contrastive learning is then used to align the embedded image-text pairs by optimizing an objective function. Representative VLMs, particularly CLIP cluster models Radford et al. (2021), and those developed on BLIP Li et al. (2022) and ALIGN Jia et al. (2021), are pre-trained on large-scale VL datasets. Another large group are Multimodal LLM (MLLM), which employs pre-trained LLM as backbone Yin et al. (2024). The visual features are extracted by a vision encoder, projected into a shared embedding space of text tokens, and inserted into text sequence as input to LLM to predict next tokens auto-regressively. Recent MLLMs are LLaVA Liu et al. (2023), GPT-4V Yang et al. (2023), Gemini Gemini Team (2025), InternVL Chen et al. (2024), and Qwen-VL Bai et al. (2023b). In this study, we employ EvaCLIP and InternVL2(8b) as representative VLM and MLLM, respectively. These models are less than 10B, and are more portable and practical for downstream applications.

**VLC benchmarks.** The benchmarks for evaluating the capabilities of latest LVLM on VLC are growing rapidly Li et al. (2025b); Fu et al. (2024). VLC capabilities are mostly evaluated using Visual Question Answering (VQA) and Referring Expression Comprehension (REC) or Visual Grounding. Traditional VQA benchmarks typically involved multiple choice formats or answers with limited text length. Subsequent VL benchmarks have been extended to include a wider range of question types, field of expertise, and tasks such as math reasoning and chart understanding. The VQA datasets have also been extended to investigate bias in dataset distributions Li et al. (2024) and negations Alhamoud et al. (2025). Standard VQA datasets are considered less challenging to frontier models Li et al. (2025b) due to vast and extended pre-training datasets. We perform experiments on a latest benchmark with negations to evaluate the effectiveness on zero-shot setting. The REC task aims to identify and localize a specific image region of the referred object on text description Xiao et al. (2024). RefCOCO/+/g are standard datasets for REC, built on top of MS COCO dataset. RefCOCO/+ might not be adequate for evaluating LVLM due to their limitations on overly concise referring phrases and limited vocabulary. On the other hand, zero-shot REC is still a very challenging task Han et al. (2024). In this study, we conduct experiments on HC-RefCOCOg with enriched phrase of 8.9 words in average, wrt 3.3 and 3.4 of RefCOCO/+, and a recent challenge HC-RefLoCo with long context expression of 93 words in average Wei et al. (2024).

**Logic reasoning on LLMs and VLM.** Efforts to enhance LLM with logical reasoning abilities have attracted researches' attention Cheng et al. (2025); Pournemat et al. (2025). The goal is to improve the logical accuracy and consistency of the generated text answers and expressions. The approaches include introducing an external logic solver Olausson et al. (2023); Diego et al. (2025) or building

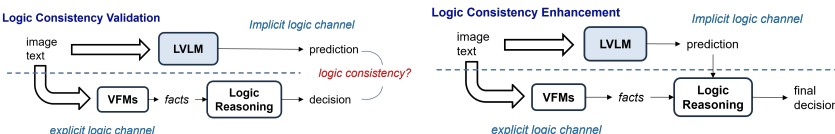

Figure 1: The Logic Channel for validation (left) and enhancement (right).

additional dataset with logically consistent annotations to fine-tune a model Feng et al. (2024). In BIRD Feng et al. (2025), probabilistic inference is used to improve the trustworthiness of LLM. Existing NeuroSymbolic framework and programming require additional learning to perform logic reasoning Li et al. (2023); Huang et al. (2025); Yang et al. (2024).

## 3 METHODOLOGY

Typical VLC tasks could be: (A) given a query image and a text expression to predict a right answer, or (B) given a query image and a text prompt to localize a target object in the image. They can be viewed as a logic problem of *if I (image) and T (text), then D (decision)*. We propose a Logic Channel in parallel with LVLM, as shown in Figure 1. The upper channel employs a LVLM to make prediction directly as a black box, which can be viewed as an *Implicit Logic Channel*. The lower channel, mimicking human logic reasoning, employs LLM and VFM to ground the facts explicitly, and performs logic reasoning on the grounded facts for final decision, which can be considered as an *Explicit Logic Channel*.

The two channels are complementary as LVLM, LLM and VFM are trained on different purposes. ELC provides explicit facts and relations to support the prediction. The logic consistency between the two channels can be used for model validation and selection, while the combination of the two predictions would further enhance the accuracy of final prediction.

**Validation**: When applying a LVLM for a new task, we may just have some test examples without gt annotation. We can run the model with Logic Channel to evaluate the logic consistency between the two channels. We define a Consistency Rate ($CR$) as

$$CR = \frac{\text{Number of Aligned Predictions}}{\text{Total Number of Test Samples}}, \tag{1}$$

where an aligned prediction means that the both channels give the same prediction. Naturally, if $CR$ is higher, the LVLM is more logically reliable and trustworthy for the new task. The metric $CR$ can be used to select a most suitable model for a new VLC task even the gt annotations are not ready for evaluation. The difference between the two channels may also guide user to perform a manual validation effectively on grounded visual evidence.

**Enhancement**: Fusing the outputs of the two complementary channels could also enhance the accuracy of the final prediction without additional re-training or fine-tuning, and the explicit visual evidence from ELC can provide a concrete logic justification of the prediction.

In this study, we investigate the effectiveness of Logic Channel with three general logic capabilities on VLC tasks, *i.e.*, (a) logical decision on factual and counter-factual evidence, (b) logical decision on evidence of facts and relations, and (c) logical decision on conditions of causation. We propose three LC solutions on these VLC requirements and evaluate their effectiveness on three recent VLC challenges on zero-shot setting.

### 3.1 LOGICAL DECISION ON FACTS AND NEGATIONS

A typical task of vision-language understanding is that, given an image ($I$), the model is required to predict a correct text description ($T$). A latest challenging benchmark, NegBench Alhamoud et al. (2025), involves both positive and negative phrases. Four choices of text descriptions are provided for each image. The choices are classified into three linguistic templates: Affirmation, Negation, and Hybrid. An affirmation text contains only positive elements $\{pos\}$, i.e., objects or concepts present in the image. A negation text includes only negative elements $\{neg\}$, which are absent from the image but commonly associated with the present objects. A hybrid text contains both positive and

negative elements. The SOTA performance shows the difficulty of existing LVLMs to understand both positive and negative descriptions on a query image.

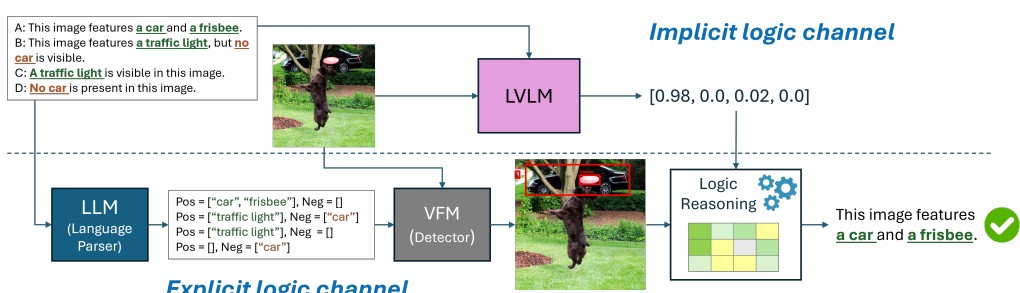

Figure 2: Logic Channel for NegBench.

Logically, the NegBench task requires decision to be made and justified on concrete factual and counter-factual evidence, i.e., the presence of pos elements (facts) and absence of neg elements (counter-facts). We implement a LC solution for the MCQ task of the NegBench benchmark, as illustrated in Figure 2. A representative LVLM is employed for the ILC. Following the protocol in Alhamoud et al. (2025), for each problem, the query image is presented to the model along-side four choice texts. The model is prompted to select the correct answer as well as provide the confidence-level of correctness (0%-100%) for all four choices. In the ELC, first, a SOTA LLM is employed as the Language Parser to extract the pos and neg nouns. The nouns are then fed to a VFM. For each object category, the VFM locates all instances in the image with the probabilities of their presence. The probability of an object category is obtained as the maximum of all the instances. Assuming there are $K$ pos object categories and $L$ neg object categories, the presence of the object categories in the image are denoted as $\{P(O_p^k)\}_{k=1}^K$ and $\{P(O_n^l)\}_{l=1}^L$. The presence of pos objects gives a factual evidence on $T$ and the presence of neg objects gives a counter-factual evidence on $T$ can be computed logically as

$$P(pos) = \min\{P(O_p^1), \cdots, P(O_p^K)\}, \quad P(neg) = \max\{P(O_n^1), \cdots, P(O_n^L)\}. \tag{2}$$

For a pair of image $I$ and text $T$, the probability of factual and counter-factual evidence can be computed as

$$P(T|I) = \begin{cases} P(pos), & \text{affirmation} \\ 1 - P(neg), & \text{negation} \\ [P(pos)(1 - P(neg))]^{\frac{1}{2}}, & \text{hybrid} \end{cases} \tag{3}$$

where the geometric mean is used for normalization for the hybrid case. The choice of maximum $P(T|I)$, out of the four given choices, is selected as the correct answer.

When performing logic consistency validation, the two channels are run independently, and the $CR$ is then computed. For enhancement, the combined probability is obtained as the sum of the two channels' outputs, and the best choice is selected as the final prediction.

## 3.2 LOGICAL DECISION ON FACTS AND RELATIONS

Recent HC-REC (Human-Centric Referring Expression Comprehension) benchmarks, sourced from general REC task, are created for the evaluation of large multi-modal models on VLC capabilities Wei et al. (2024). Given a query image ($I$) and a text expression ($T$), the model is required to predict and localize the referred person ($H$) within the image. The prediction should be made on the presence of the person and relevant objects, as well as the evidence of their relations in the image. The presence of a person is the core factual evidence, and his/her relations with the mentioned objects provide the evidence of association on logic relations for prediction.

The Logic Channel solution for HC-REC is shown in Figure 3. In the ILC, we first employ a VFM to locate every person in the image. Then, each person is cropped, and fed to a LVLM along with the referring expression. The LVLM makes a prediction on each person with a confidence value on its learned VLC knowledge. In the ELC, the Language Parser is prompted to extract the nouns of

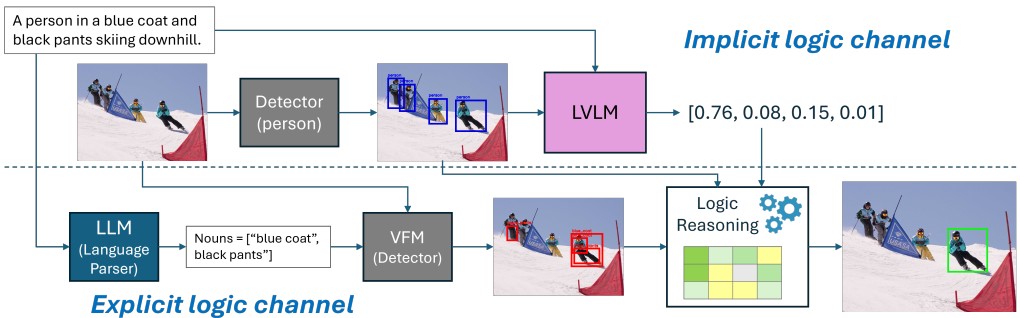

Figure 3: Logic Channel for HC-REC task.

persons and objects from $T$. The nouns are then fed to a VFM to locate all instances of persons and relevant objects with the probabilities of their presence in the image.

Assuming that there are $(K + 1)$ nouns extracted from $T$, of which the first is human and the rest are object names, denoted as $\{H, O_1, \cdots, O_K\}$. The VFM locates a set of instances of human and objects. There may be multiple instances of one object category. In addition, due to errors of zero-shot detection, there may be missed detection and false positives. Let us denote the box of the $i$th detected person as $h_i$, and the box of the $l$th instance of object $O_j$ as $o_{jl}$.

We compute the probability of the targeted person on accumulated evidence, as inspired by the principle of perceptual decision-making on evidence accumulation Balsdon et al. (2020). Formally, it can be expressed as

$$P(h_i|T) = \frac{1}{K+1}\left(\sum\nolimits_{k=1}^{K} P(O_k|h_i) + P(h_i|H)\right), \tag{4}$$

where $P(h_i|H)$ represents the probability of a person present at $h_i$, and $P(O_k|h_i)$ denotes the association of region $h_i$ with object $O_k$. Assuming that there are $L$ instances of object $O_k$ detected in the image, i.e., $\{o_{kl}\}_{l=1}^{L}$, then, the association rate of $o_{kl}$ with $h_i$ can be computed as

$$R_A(o_{kl}, h_i) = A_{int}(o_{kl}, h_i)/A(o_{kl}), \tag{5}$$

where $A_{int}(o_{kl}, h_i)$ means the area of intersection of boxes $o_{kl}$ and $h_i$, and $A(o_{kl})$ is the area of box $o_{kl}$. The probability of association of $h_i$ with an object named $O_k$ is obtained as

$$P(O_k|h_i) = \max_{l\in[1,L]}\{R_A(o_{kl}, h_i)\}. \tag{6}$$

Finally, the person of maximum $P(h_i|T)$ is selected as the grounded person referred by text expression on explicit visual evidence.

For model validation, the two channels are run independently. In ELC, $P(h_i|H) = 1.0$ is used in (4), which means every detected person is assumed as a potential target person. Then, the $CR$ is computed. For enhancement, the confidence value of LVLM's prediction of $h_i$ on $T$ is used as $P(h_i|H)$ in (4), which integrates the power of LVLM in logic reasoning.

### 3.3 LOGICAL DECISION ON CONDITIONS OF CAUSATION

A recent challenge of REC is HC-RefLoCo, *i.e.*, Human-Centric Referring Expression Comprehension with Long Context Wei et al. (2024). The very long text expression, which contains multiple sentences about the scene, global events and activities, to individual appearance and action, cause difficulty and even failure of many LVLMs for zero-shot task.

The challenges are not simply caused by long context expression, but also by sentences that are not logically complete for the cause-effect prediction. On logic cause-effect relation Gomes (2024), given image $I$, the text $T$ should be sufficient and necessary to cause the localization of $H$ in $I$. However, many sentences in $T$ are neither sufficient nor necessary to lead to the visual attention to $H$ in $I$. This can be explained in the following example. Suppose a long context expression on an image of human activities in a public park in a weekend is given as: "*This is a scene of a public park*

*in Sunday. People play various activities in the morning. A young lady in red sportswear is sitting on a bench for a rest. A blue bottle is placed on the bench beside her.*" The first sentence ($S_1$) is neither sufficient nor necessary to lead to the attention to the young lady ($H$). The second ($S_2$) is a sufficient but not necessary cause to $H$ since it may lead to attention to every person in $I$, not just $H$. The third ($S_3$) is a sufficient and necessary cause to $H$, if there is no other young lady in red sportswear sitting on a bench in $I$. The fourth ($S_4$) is necessary but may not be sufficient cause as it does not focus on person.

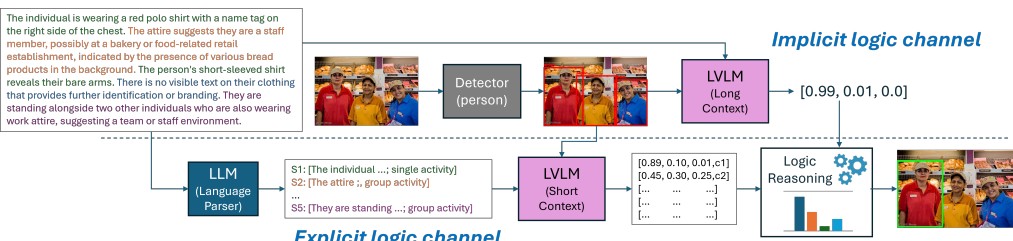

Figure 4: Logic Channel for HC-RefLoCo.

A LC solution for HC-RefLoCo is developed, as shown in Figure 4. On this challenge, we focus on the effectiveness of understanding the logic causation conditions on long context expressions with LVLMs. We employ LVLMs in both channels, one works on the long text of full expression, and another works on short text of sentence. The persons detected by VFM are used in both channels as potential visual grounds. In ILC, each person is cropped from the image, and fed to a LVLM along with the full expression. The LVLM produces predictions on its learned VLC knowledge, each with a confidence value of its prediction. In ELC, a logic reasoning approach on LLM and LVLM is developed. First, each sentence is extracted from the full expression and a LLM is prompted to classify each sentence into four categories: single activity (e.g., $S_3$), object (e.g., $S_4$), group activity (e.g., $S_2$) and scene (e.g., $S_1$). On logic formation, they could be assumed to correspond to different logic conditions of causation, *i.e.*, (a) Single Activity: a sufficient and necessary cause; (b) Object: a necessary but not sufficient cause; (c) Group Activity: a sufficient but not necessary cause; and (d) Scene: a neither sufficient nor necessary cause.

Then, for a detected person, the cropped patch and each sentence form a VL pair. It is fed to a lightweight LVLM. Hence, for each person and each sentence, we obtain a probability of LVLM's prediction on the VL association. If there are $N$ persons detected and $K$ sentences extracted from $T$, the predicted $HS$ matches can be denoted as $\{\{P_{LVLM}(h_n|S_k)\}_{k=1}^K\}_{n=1}^N$. On the principles of probabilistic logic causation Gomes (2024), probabilistic inference framework on factors Feng et al. (2025), and human decision-making theory Bradley (2018), the logical prediction is obtained as

$$P(h_n|T) = \sum_{k=1}^{K} P_{LVLM}(h_n|S_k)u(S_k) \tag{7}$$

where $u(S_k)$ can be considered as a utility weight of $S_k$ for perceptual decision-making, which can be selected on sentence category (see Sec. Experiments and Supplementary Material). The $h_n$ of maximum $P(h_n|T)$ is selected as $H$.

Different $LVLM$s can be used for the Implicit and Explicit Logic Channels, on their performance on long/short text expressions. For model validation, the two channels are run independently, and the results are compared for logic validation and model selection. For enhancement, a simple weighted sum can further improve the accuracy of the final prediction.

## 4    EXPERIMENTS

In this section, we present the quantitative results and analysis on the effectiveness of Logic Channel for model validation, justification, selection, and enhancement of prediction when applying LVLM to challenging VLC tasks in zero-shot setting. In our experiments, we employ two LVLMs, i.e., EvaCLIP and InternVL2(8b), as representative models of VLM and MLLM, respectively. The LLM used as Language Parser is Mistral Jiang et al. (2023), and the VFM for person and object detection is GroundingDINO Liu et al. (2024). In the following, ILC-LVLM denotes the results from ILC,

and ELC-LVLM for the results from ELC, where LVLM$\in$ {EvaCLIP, InternVL2(8b)}, referring to the corresponding LVLM used in the channel.

### 4.1 EVALUATIONS ON NEGBENCH BENCHMARK

NegBench Alhamoud et al. (2025) is one of most recent VLC challenges. There are three natural image MCQ tasks, i.e., COCO, VOC2007, and HardNeg-Syn. Our experiments are conducted on the publicly available data, i.e., 5914 MCQ questions on COCO and 5032 MCQ questions on VOC2007. The SOTA accuracies are extracted from Table 2 and Figure 7 in the Appendix of the paper Alhamoud et al. (2025).

**Validation.** First, we evaluate the $CR$ without gt annotation. The results of the ILC on EvaCLIP and InternVL2(8b) are presented in Table 1. The $CR$ of EvaCLIP is around 27%, slightly over 25% of random chance. Even without gt annotation, one can predict that EvaCLIP is poor for the task, according to the logic validation on the factual and counter-factual evidence. On the other hand, the $CR$ of InternVL2(8b) is around 70% across the two datasets, much higher than EvaCLIP, which provides a logic validation that InternVL2(8b) may be more suitable for the task.

| Benchmark | Dataset | EvaCLIP | InternVL2(8b) |
|-----------|---------|---------|---------------|
| | COCO | 27.05 | 62.80 |
| NegBench | VOC | 27.25 | 79.25 |
| | ALL | 27.14 | 70.36 |

Table 1: Logic Consistency Rates on NegBench.

| Model | COCO | VOC2007 |
|-------|------|---------|
| CLIP-Laion400M (2022) | 24.26% | 27.01% |
| CLIP-L14(2021) | 22.44% | 23.69% |
| CLIP-H14(2021) | 32.14% | 38.26% |
| LLaVA(7b)(2023) | 48.00% | 56.00% |
| LLaVA(13b)(2023) | 54.00% | 63.00% |
| ILC-EvaCLIP | 24.82% | 26.60% |
| ILC-InternVL2(8b) | 63.26% | 85.41% |
| ELC | **75.77%** | 86.84% |
| ILC-InternVL2+ELC | 70.34% | **88.25%** |

Table 2: MCQ Total Accuracy (%) on NegBench.

| Model | RefCOCOg | |
|-------|----------|------|
| | Val | Test |
| ReCLIP(2022) | 59.33 | 59.01 |
| RelVLA(2024) | 57.60 | 56.64 |
| GroundVLP(2024) | 64.30 | 63.54 |

| Model | HC-RefCOCOg | |
|-------|-------------|------|
| | Val | Test |
| ReCLIP(2022) | 68.36 | 67.33 |
| RelVLA(2024) | 65.65 | 64.57 |
| GroundVLP(2024) | 74.44 | 74.81 |
| ILC-EvaCLIP | 78.40 | 78.66 |
| ILC-InternVL2(8b) | 47.80 | 51.66 |
| ELC | 67.68 | 67.52 |
| ILC-EvaCLIP+ELC | **80.70** | **80.05** |

Table 3: Performance evaluation on HC-RefCOCOg.

When the gt annotations are used for performance evaluation, one can observe the results of individual ILC or ELC in Table 2. The accuracy of EvaCLIP is around 25%, agreeing with $CR$ validation and the results of CLIP cluster models reported in Alhamoud et al. (2025). As indicated by $CR$ validation, InternVL2(8b) produces much better performance than EvaCLIP, and achieves a large jump over the SOTA by LLaVA(13b), one of the MLLMs.

**Enhancement.** When we simply sum up the predictions from both ILC and ELC, we obtain performance better than the end-to-end model on COCO, but lower than the best single channel due to a large gap between the two channels. Nonetheless, we obtain further improvement on VOC2007, increasing SOTA accuracy from 63% to over 88%.

### 4.2 EVALUATIONS ON HC-REFCOCOG BENCHMARK

In this experiment, we focus on HC-RefCOCOg with enriched phrase. As there is still a lack of benchmarks on HC-RefCOCOg, we download publicly available SOTA models on RefCOCOg and run on HC-RefCOCOg. In Table 3, to compare with SOTA of zero-shot REC, we list the published results on RefCOCOg, as well as the corresponding results on HC-RefCOCOg obtained by us.

**Validation.** First, we evaluate the logic consistency on the val set without gt annotation. The $CR$ on EvaCLIP and InternVL2(8b) are 63.1% and 40.8% respectively. The former is around 20% higher than the latter, which may indicate that EvaCLIP may be more suitable for the task than InternVL2.

When the gt annotations are used for performance evaluation, one can observe the results of individual ILC with EvaCLIP and InternVL2(8b) in Table 3. As indicated by $CR$, the performance

of EvaCLIP is more than 20% over InternVL2, and achieves an improvement of around 4% over the SOTA. The ELC obtains performance of over 67%, close to the performance of RelVLA and ReCLIP, indicating that it can be used as an effective and reliable tool for model validation with explicit visual evidence.

**Enhancement.** By introducing the predictions of ILC into Eq. (4), the integrated LCR model (last row) produces new SOTA performance of over 80%, around 6∼15% over previous SOTA and 1.8% over ILC with EvaCLIP, demonstrating the effectiveness of Logic Channel for enhancing the accuracy of the final prediction.

### 4.3 EVALUATIONS ON HC-REFLOCO BENCHMARK

The recent challenge HC-RefLoCo Wei et al. (2024) is built on various image datasets, e.g., COCO, Objects365, OpenImage v7, and LAION-5B. On a query image, the long context annotation is generated by GPT-4V with user prompts and reviewed manually. The long text expressions form a great challenge to most LVLMs not pre-trained on long texts. Here, we show that it is possible to develop a LC solution with lightweight LVLMs and FMs (Figure 4) for the HC-RefLoCo challenge.

| Model | Val+Test | | | | Val | Test |
|---|---|---|---|---|---|---|
| | $Acc_{0.5}$ | $Acc_{0.75}$ | $Acc_{0.9}$ | mAcc | mAcc | mAcc |
| Qwen-VL Bai et al. (2023a) | 67.9 | 56.8 | 34.8 | 52.8 | 53.1 | 52.6 |
| CogVLM Wang et al. (2024) | 66.0 | 59.6 | 43.8 | 55.8 | 56.3 | 55.5 |
| ONE PEACE Wang et al. (2023) | 79.3 | 69.0 | 43.8 | 63.1 | 63.4 | 62.9 |
| SPHINX-MoE-1k Lin et al. (2023) | **85.8** | 77.3 | 53.7 | 71.4 | 71.5 | 71.4 |
| SPHINX-v2-1k Lin et al. (2023) | 84.1 | 77.1 | 56.2 | 71.7 | 71.6 | 71.7 |
| ILC-EvaCLIP | 75.3 | 70.5 | 59.1 | 67.6 | 67.2 | 67.8 |
| ILC-InternVL2(8b) | 33.5 | 31.4 | 24.9 | 29.6 | 29.1 | 29.8 |
| ELC-EvaCLIP | 67.7 | 63.4 | 52.5 | 60.6 | 60.2 | 60.8 |
| ELC-InternVL2(8b) | 80.8 | 75.9 | 63.4 | 72.7 | 71.8 | 73.1 |
| ILC-EvaCLIP+ELC-InternVL2 | 83.4 | **78.3** | **65.2** | **75.0** | **74.5** | **75.2** |

Table 4: Performance evaluation on HC-RefLoCo benchmark.

**Validation.** First, we evaluate if a LVLM is effective on long context expressions on the logic consistency without gt annotation. We use EvaCLIP or InternVL2(8b) in both the ILC and ELC respectively. In the ELC, we simply sum up the probabilities on sentences. The $CR$ between the two channels on EvaCLIP and InternVL(8b) are 72.8% and 15.8% respectively. The results indicate that, EvaCLIP might be less sensitive to the truncated long context expressions, while InternVL2(8b) might be too sensitive to the truncation of long context expressions.

The ILC with EvaCLIP and InternVL2(8b) are evaluated directly on the full expressions. The results are listed in Table 4 (i.e., ILC-EvaCLIP and ILC-InternVL2(8b)). Both LVLMs may truncate the long text input. However, in EvaCLIP, the truncated text input is embedded as a text vector of a fixed length and compared with the visual feature vector embedded as the same length. The result shows that EvaCLIP still lags behind the SOTA (SPHINX) but it achieves better result of $Acc_{0.9}$ than SOTA, with the localization by GroundingDINO. On the other hand, InternVL2(8b) interprets the input text tokens one-by-one for next prediction, the truncation of the long text input causes poor performance on the HC-RefLoCo task.

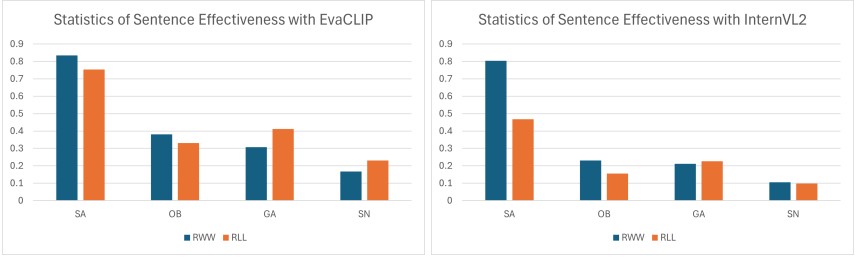

Figure 5: The statistics of $R_{WW}$ and $R_{LL}$ on sentence categories on HC-RefLoCo val set.

In the ELC, a full expression is first separated as sentences, and the sentences are classified into four categories, labeled as SA, OB, GA and SN, denoting single activity, object, group activity, and scene respectively. Each sentence represents a degree of necessity or sufficiency, or a probability of the causation. Suppose there are $K$ sentences and $N$ persons in the image. Each sentence, together with all patches of cropped persons, are fed to a LVLM, and it is prompted to find the best match, with confidence values for its predictions. All together, we obtain LVLM predictions $\{\{P_{LVLM}^{n,k}\}_{k=1}^{K}\}_{n=1}^{N}$. If each sentence provides appropriate information and the LVLM is able to understand the VL information correctly, when we sum up the probabilities over all sentences for each person, the gt person would have the maximum probability, or labeled as 'GT-Win'. For each person, we can also sum up the probabilities of each category, e.g., $P_{SA}^{n}$. If $P_{SA}^{n}$ of the gt person is also the maximum of SA cluster (denoted as 'SA-Win'), SA contributes to GT-Win. We compute a Rate of Win-Win for SA and a Rate of Lose-Lose for SA as

$$R_{WW}(SA) = \frac{Num(\text{SA-Win\&GT-Win})}{Num(\text{GT-Win})}, \quad R_{LL}(SA) = \frac{Num(\text{SA-Lose\&GT-Lose})}{Num(\text{GT-Lose})}. \tag{8}$$

In the same way, we can obtain the Rates for OB, GA, and SN categories. The statistics of the Rates with EvaCLIP and InternVL2(8b) on val set are shown in Figure 5. Logically, if SA-Win and GT-Win occur, we may view SA-Win as a sufficient condition of GT-Win, i.e., *if SA-Win, then GT-Win*. On the other hand, if SA-Lose and GT-Lose occur, we may view SA-Win is a necessary condition for GT-Win, i.e., *if not-SA-Win, then not-GT-Win*, or *if SA-Lose, then GT-Lose*. When examining the bars in Figure 5, one can observe that SA has strong relation with gt, while OB has weaker relation with gt, but still has more positive effect ($R_{WW}$) than negative effect ($R_{LL}$), and the remaining two categories have less relation with gt. On such statistics, we obtain the utility weights of SA, OB, GA, and SN (see Supplementary Material), and apply them in Eq. (7) for logic prediction.

The results of Logic Channel on sentences using EvaCLIP and InternVL2(8b) in ELC are shown in Table 4 (i.e., ELC-EvaCLIP and ELC-InternVL(8b)). The performance of ELC-EvaCLIP is poorer than ILC-EvaCLIP on full annotation, maybe due to missing information in separated sentences, but it is still comparable with most SOTA models except SPHINX. ELC-InternVL2(8b) produces much better performance on sentences, generates new SOTA results except $Acc_{0.5}$, especially, increases $Acc_{0.9}$ from 56.2% to 63.4%, significantly improving the accuracy of localization.

**Enhancement.** The model validation results show that EvaCLIP is effective on full annotation, and InternVL2(8b) is effective on sentences. By combining the two channels with a weighted sum, we obtain new SOTA performance except $ACC_{0.5}$, improving the mAcc from 71.7% to 75.0%, $Acc_{0.75}$ from 77.3% to 78.3%, and $Acc_{0.9}$ from 56.2% to 65.2%.

### 4.4 VISUAL VALIDATION AND JUSTIFICATION

One critical limitation of LVLM is the lack of explicit explanation of its decision making. The uncertainty may be caused by visual hallucination or ambiguity of gt annotation. With Logic Channel validation, we may effectively address such concerns by manually examining the inconsistent results between LVLM and ELC even without gt annotation. The inconsistencies may be caused by: (a) Hallucination: LVLM makes a prediction on nonexistent object or fails to see the referred object; (b) Ambiguity: LVLM makes a decision on neither sufficient nor necessary condition, or gt annotation is incorrect; or (c) Weakness of FMs: LLM fails to extract correct concept, or VFM fails to detect the relevant objects. The visual examples of such errors observed on explicit visual evidence in logic reasoning are presented in the Supplementary Material with further discussions.

## 5 CONCLUSION

Facing the uncertainty on reliability when deploying frontier LVLM to new tasks, we proposed a Logical Channel for model validation and enhancement w/o re-training or fine-tuning. Based on Foundation Models and logic reasoning, the Logic Channel make decision on explicit facts and relations, which provides a way for model validation on human logic. We investigated the effectiveness on three recent benchmark challenges in VLC. This Logic Channel enables diagnosis for model validation, selection and improvement, as well as enhancement of explainability and trustworthiness. The diversity of the three challenges demonstrates the generalization of our approach. In the future, it is worth to investigate the LC for validation of more complex multimodal CoT reasoning tasks.

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
