# SUPPLEMENTARY MATERIAL

## 1 CONCEPTUAL BLOCK DIAGRAMS

We propose an Explicit Logic Channel inspired by human logic reasoning as logic validation of LVLM performance without learning and gt annotation. We provide concept-level block diagrams for better overview of concepts and process flow.

### 1.1 MOTIVATION OF ILC AND ELC FRAMEWORK

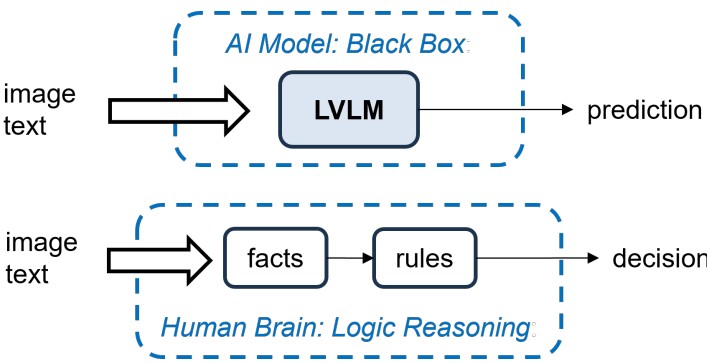

Figure 1: The motivation of Explicit Logic Channel and Logic Consistency Validation by comparing LVLM (Implicit Logic Channel) prediction and human brain reasoning.

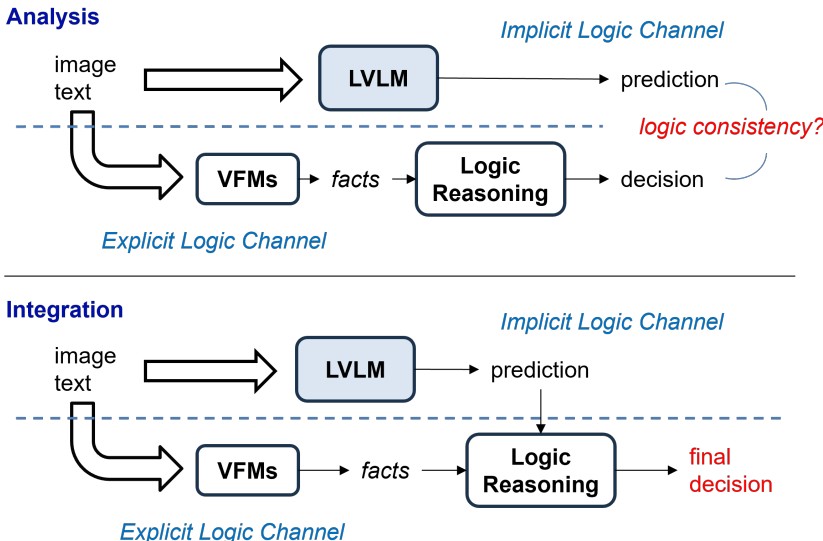

Figure 2: The proposed General Framework with ILC and ELC for validation (upper) and enhancement (lower).

The motivation of Explicit Logic Channel and Logic Consistency validation is illustrated in Figure 1, where the processes of the AI system on LVLM and human cognition for visual-language

comprehension (VLC) are compared in parallel. This comparison inspires us to propose a dual-channel framework shown in Figure 2, where the upper Implicit Logic Channel (ILC) represents the AI Model process, and the lower Explicit Logic Channel (ELC) mimics the human cognition process for VLC.

## 1.2 ELC FOR MCQ ON FACTS AND NEGATIONS

The concise block diagram of ELC for MCQ task of NegBench is shown in Figure 3. The diagram combines both validation and enhancement procedures. In Validation procedure, there is no connection from ILC to ELC on the right side. The predictions from ILC and ELC are obtained separately and compared for logic consistency validation. In the Enhancement procedure, the predictions from ILC are combined in ELC for the final prediction.

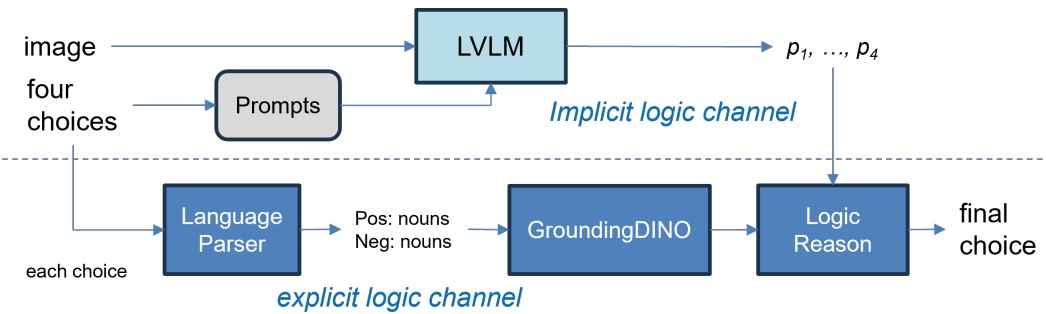

Figure 3: ELC for MCQ task in NegBench.

In ILC, following the standard MCQA format, the query image and four answer choices are provided to the LVLM, where it is prompted to select the right answer. Due to the requirement of forced to select one answer from four choices, this might lead to shortcut learning in LVLM. In ELC, we follow the natural VLC procedure. The query image and each text choice are fed to ELC, where ELC makes predictions simply on the visual-language understanding, i.e., the presence and/or absence of relevant objects, without constraint to select one from four choices.

## 1.3 ELC FOR HC-REC ON FACTS AND RELATIONS

The concise block diagram of ELC for HC-REC task on facts and relations referred in text phrase, experimented on HC-RefCOCOg benchmark, is shown in Figure 4. The diagram combines both Validation and Enhancement procedures. In Validation procedure, there is no connection from ILC to ELC in the right side. The predictions from ILC and ELC are obtained separately and compared for logic consistency validation. In the Enhancement procedure, the predictions from ILC are combined in ELC for the final prediction.

For the task of Human-Centric REC, the presence and location of a person becomes a sufficient condition to localize the target person referred by the text in the query image. Hence, GroundingDINO is employed for human detection in both ILC and ELC, which also provide a condition for fair comparison between ILC and ELC.

## 1.4 ELC FOR HC-REC ON CAUSATION CONDITIONS

The concise block diagram of ELC for HC-REC task on causation conditions extracted from long context text annotation, experimented on HC-RefLoCo benchmark, is shown in Figure 5. The diagram combines both Validation and Enhancement procedures. In Validation procedure, there is no connection from ILC to ELC in the right side. The predictions from ILC and ELC on a same LVLM are obtained separately and compared for logic consistency validation. In the Enhancement procedure, the predictions from ILC are combined in ELC for the final prediction, where, in ILC and ELC, different LVLM models can be employed for better performance.

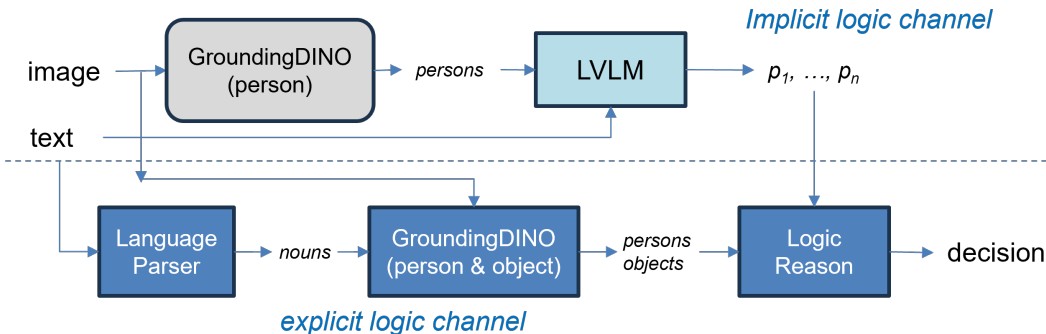

Figure 4: ELC for HC-REC task on facts and relations referred in a short text phrase, experimented on HC-RefCOCOg benchmark.

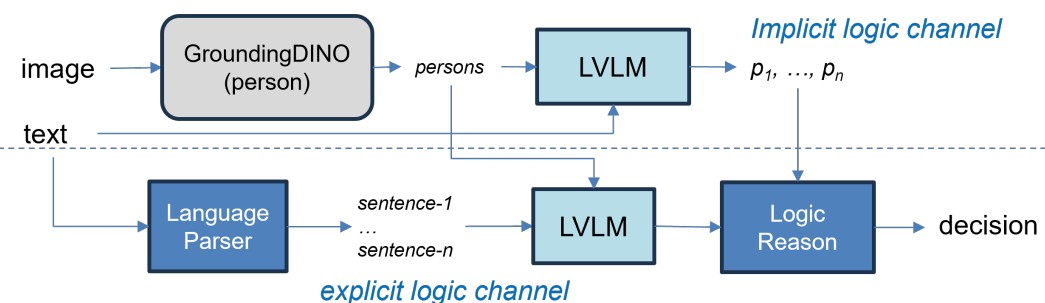

Figure 5: ELC for HC-REC task on long context annotation, experimented on HC-RefLoCo benchmark.

Same as above task, for the task of Human-Centric REC, the presence and location of a person becomes a sufficient condition to localize the target person referred by the text in the query image. Hence, GroundingDINO is employed for human detection in both ILC and ELC,

On HC-RefLoCo, the long context annotation causes a challenge to LVLMs for zero-shot task, since few of light models were pre-trained on such long text inputs. On this challenge, we focus on the effectiveness of understanding the logic causation conditions on long context expressions with LVLMs, especially the light ones ($\leq$8b). Hence, LVLMs are employed in both ILC and ELC, where, the one in ILC works on the long text of full expression, and the other in ELC works on short text of sentence.

## 2 EXPERIMENTAL DETAILS

The additional experimental details are presented in this section, which include the sources of benchmark datasets used in the experiments, the system configuration, the prompts for LLM and MLLM models used in the experiments, and the parameter computing in ELC for HC-RefLoCo task.

### 2.1 LINKS OF DATA SOURCES

The three benchmark datasets can be prepared or downloaded following the links below:

- NegBench benchmark: `https://github.com/m1k2zoo/negbench`

- RefCOCOg benchmark: `https://github.com/lichengunc/refer`

- HC-RefLoCo benchmark: `https://github.com/ZhaoJingjing713/HC-RefLoCo`

## 2.2 IMPLEMENTATION DETAILS

In our experiments, we select EvaCLIP and InternVL2 as representative VLM and MLLM, respectively, Mistral as LLM Language Parser, and GroundingDINO as VFM open-set detector. For Eva-CLIP, we employ pre-trained model EVA-CLIP-8B, with image processor clip-vit-large-patch14. For InternVL2, we use the 8B version, to be comparable with EvaCLIP. For Mistral, we use Mistral-7B-Instruct-v0.2, while we use GroundingDINO with Swin-T backbone that has been pretrained on O365, GoldG and Cap4M for person and object detection. The models are implemented in a server with two Nvidia RTX2080Ti GPUs.

## 2.3 PROMPTS FOR LLM

**NegBench**: Mistral is used for for language parsing for NegBench. The prompt for extracting present and absent nouns is as follows:
*"You are given a sentence. Return a dict of present nouns and absent nouns. Example: 'No book' means 'absent': ['book'], 'cake is not visible' means 'absent': ['cake']. Note: Do not give any explanation. Output a complete JSON string with only 2 key: 'present', 'absent'."*

**REC or Visual Grounding**: Mistral is also used for language parsing for REC or Visual Grounding. The prompt for categorizing the text input and extracting object nouns, as well as other potentially useful information is presented as follows:
*"Extract the following information from the sentence: 1. Type of activity (choose one closest match): scene, group-activity, single-activity, object. 2. Number of people involved (count all people mentioned in the sentence). If the number is unknown, enter -1. If the type is 'object', enter 0. 3. Position of event (choose one closest match): 'top-left', 'top-centre', 'top-right', 'bottom-left', 'bottom-centre', 'bottom-right', 'left', 'centre', 'right', 'top', 'bottom'. If unknown, enter 'none'. 4. A list of object nouns. No person noun, no scene noun.*

*Note: 1. Do not give any explanation. 2. Make number of people a string format. 3. If position of event is 'X hand', 'X hand area', 'X side area', or 'X part', return only 'X'. 4. If position of event is 'upper-X', return only 'top-X'. 5. If position of event is 'lower-X', return only 'bottom-X'. 6. For position of event, return 'middle' and 'center' as 'centre'. 7. If sentence describes global scene, for example, 'This image describes ...', return 'scene'. 8. For object nouns, include color of object, for example, 'orange shirt', 'black shorts', 'pink guitar'. Output a complete JSON string with only five keys: 'type', 'number_of_people', 'position_of_event', 'object_nouns'."*

Spatial cues of position information are not used in the experiments due to the unreliability on complex natural language expressions. Estimated group size on text descriptions is also not used in this study as HC-REC on HC-RefCOCOg and HC-RefLoCo require to localize single person in the image.

## 2.4 PROMPTS FOR MLLM (INTERNVL2(8B))

**NegBench**: The prompt is used for InternVL2 in NegBench is presented as:
*question1 = "Is this description '+ label +' true or false? Can you also give the confidence values of the correctness on a scale of 0 to 1? The output starts with Yes or No and the confidence values using the same format.".*
The label information is obtained from the input text.

**HC-REC**: For HC-RefLoCo and RefCOCOg, the prompt is designed as
*question = "Can you give a number between 0 and 1 representing similarity of the description '+ label.rstrip('.') + ' and this image? The output is just value number using the same format."*

# 3 PARAMETER COMPUTING IN EXPERIMENTS ON HC-REFLOCO

In the experiments on HC-RefLoCo benchmark, we have mentioned that the weights of utilities on sentence categories can be obtained on the statistics of $R_{WW}$ (Rate of Win-Win) and $R_{LL}$ (Rate of Lose-Lose). Here, we present the details of the analysis. The statistics of $R_{WW}$ and $R_{LL}$ for SA, OB, GA and SN obtained when using EvaCLIP and InternVL2(8b) in ELC on val set, shown in

Figure 5 in the main paper, are presented in the top two rows in Table 1. The statistics are obtained when equal weight for all sentences is used. In Table 1, the values of % indicate the percentages of GT-Win and GT-Lose on the val set when EvaCLIP or InternVL2(8b) are employed. The third row in Table 1 shows the weighted sums of $R_{WW}$ and $R_{LL}$ for SA, OB, GA and SN categories, where the weights are the values under % column. The last row shows the normalized weights of sentence categories obtained with EvaCLIP and InternVL2(8b) models, respectively.

| | EvaCLIP | | | | | InternVL2(8b) | | | | |
|---|---|---|---|---|---|---|---|---|---|---|
| | SA | OB | GA | SN | % | SA | OB | GA | SN | % |
| $R_{WW}$ | 0.8348 | 0.3803 | 0.3067 | 0.1678 | 0.6518 | 0.8039 | 0.2304 | 0.2110 | 0.1048 | 0.7945 |
| $R_{LL}$ | 0.7541 | 0.3310 | 0.4114 | 0.2300 | 0.3482 | 0.4676 | 0.1559 | 0.2261 | 0.0976 | 0.2055 |
| Weighted Sum | 0.8067 | 0.3631 | 0.3432 | 0.1895 | | 0.7348 | 0.2150 | 0.2142 | 0.1033 | |
| Normalized | 0.4738 | 0.2134 | 0.2016 | 0.1112 | | 0.5799 | 0.1697 | 0.1690 | 0.0815 | |

Table 1: Statistics of $R_{WW}$s and $R_{LL}$s for SA, OB, GA and SN on val set of HC-RefLoCo.

To be generalized and unbiased to models, we compute the utility weights of the four sentence categories as the average of those obtained on the two models. The data are shown in Table 2. The first two rows are the normalized weights of sentence categories for EvaCLIP and InternVL2(8b), as ones shown at the bottom line of Table 1. The direct sum are shown in the third row, and the last row shows the normalized weights. The last row are the weights of utilities used in Eq. (8) and experiments in the main paper.

| | SA | OB | GA | SN |
|---|---|---|---|---|
| EvaCLIP | 0.4738 | 0.2134 | 0.2016 | 0.1112 |
| InternVL2(8b) | 0.5799 | 0.1697 | 0.1690 | 0.0815 |
| Sum | 1.0537 | 0.3831 | 0.3706 | 0.1927 |
| Normalized | 0.5268 | 0.1915 | 0.1853 | 0.0963 |

Table 2: Average weights on val set of HC-RefLoCo.

# 4 VISUAL VALIDATION AND JUSTIFICATION

One critical limitation of LVLMs is the uncertainty of their prediction due to lack of explanation. With explicit logic reasoning, we may effectively address such concern by manually examining the results of ILC and ELC even without gt annotation. We can check the test samples with inconsistent predictions by ILC and ELC. The inconsistencies may be caused by:

- Hallucination: LVLM makes a prediction on nonexistent object or fails to see the referred object;

- Ambiguity: LVLM makes a decision on neither sufficient nor necessary condition, or gt annotation is incorrect;

- Weakness of FMs: LLM fails to extract correct concept, or VFM fails to detect the relevant objects.

The visual examples of such errors observed on explicit visual evidence in logic reasoning are presented in the following.

## 4.1 EXAMPLES FROM NEGBENCH

Four examples from NegBench are presented in Figure 6, where ILC on InternVL2(8b) and ELC on factual and counterfactual reasoning produce inconsistent predictions. In each image, the red box and text indicates a pos object, and the blue box and text indicates a neg object. The text at the top of the image indicates that the referred object is not detected, with red text referring to pos object and blue text for neg object. Under each image, the four text choices are presented. Below them, the blue texts present the GT choice, the choices predicted by ILC and ELC, respectively. At the bottom, the red text indicates the cause of the error.

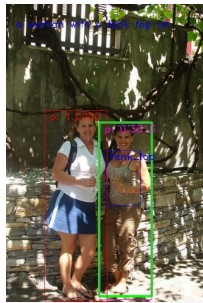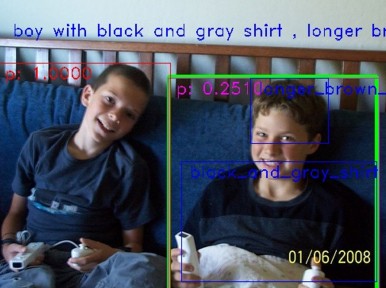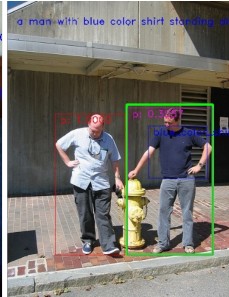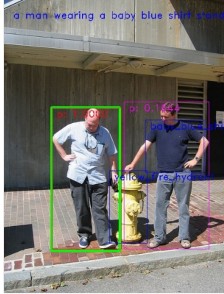

1 This image features a carrot but no bottle.
2 This image contains a bottle but no carrot.
3 A bottle is included in this image.
4 No carrot is included in this image.
**GT: 1, ILC: 1, ELC: 4.**
*Hallucination on nonexistent object 'carrot'.*

1 A car is present in this image, but there is no bus.
2 A bus is present, but there is no car.
3 This image shows a bus.
4 No car is in this image.
**GT: 1, ILC: 1, ELC: 3.**
*Hallucination on missed object 'bus'.*

1 A dining table is not present in this image.
2 This image features a dining table, but it does not include a chair.
3 This image features a dining table.
4 No chair is visible in this image.
**GT: 1, ILC: 2, ELC: 3.**
*Ambiguous GT object 'dining table'.*

1 A mouse is present in this image, but there is no dining table.
2 This image shows a dining table, with no mouse in sight.
3 A dining table is featured in this image.
4 There is no mouse in this image.
**GT: 1, ILC: 1, ELC: 3.**
*VFM error on missed detecting 'writing table' as 'dining table'..*

Figure 6: Examples from NegBench, where the last red text under each example indicates the cause of error on explicit visual evidence.

a woman with a tank top on

boy with black and gray shirt , longer brown hair

a man with blue color shirt standing along with him friend

a man wearing a baby blue shirt standing next to a yellow fire hydrant

Green box: GT, Red box: ILC prediction, Pink box: ELC prediction, Blue box with blue text: referred object associated with predicted person by logic reasoning.

Figure 7: Examples from HC-RefCOCOg.

In the first example, both GT choice and ILC prediction are not correct as they are based on the hallucination of non-existent object 'carrot'. In ELC, the VFM (GroundingDINO) does not detect carrot on the table, leading to counterfactual reasoning and selecting the right choice. In the second image, both bus and car are partially occluded. Again, both GT choice and ILC prediction are not correct as they missed the presence of the partially occluded bus. However, in ELC, the VFM (GroundingDINO) detects the partially occluded bus in the left, which leads to logic reasoning on factual evidence to make a right choice. In the third example, the ambiguous concept of 'dining table' may cause the different choices. In the image, a man is using the small table at the back of the seat in front as dining table. On GT choice, the small table is not considered as dining table by human annotator. In ELC, GroundingDINO detect it as a dining table, leading to the 3rd choice. The explicit indication of the pos object, i.e., the red box and text shown in the image, is very helpful for logic justification of its choice. In the fourth example, it is observed that GroundingDINO considers the desk as a dining table when it is asked to detect a dining table. Such error may cause the wrong prediction of ELC. Again, the explicit indication of pos object as factual evidence makes it easy for logic validation of the answer.

## 4.2 EXAMPLES FROM HC-REFCOCOG

Four examples from HC-RefCOCOg are presented in Figure 7, where ILC on EvaCLIP and ELC on factual and relational reasoning produce inconsistent predictions. In each image, the green box indicates the GT box, the red box and text indicate the prediction made by ILC with the probability

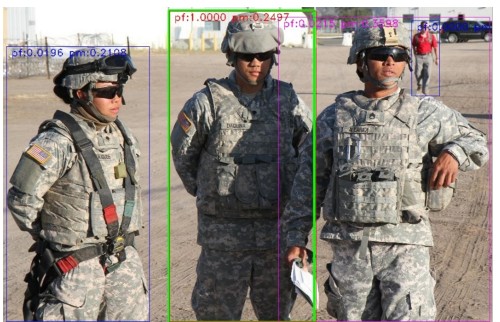

Full annotation 2:
The individual in the image is dressed in a U.S. military uniform, which features a digital camouflage pattern. The uniform is adorned with various patches, including the American flag on the right sleeve. The person is wearing a helmet and is also equipped with a tactical vest, which has pouches and gear attached to it. On the chest, there is a name tape that reads "ALCARAZ". In their right hand, they are holding some papers. The person appears to be in a state of movement or walking, as indicated by the positioning of their left arm and the slight turn of their body. The location appears to be an outdoor setting, likely a military or training facility, given the attire and context.

• Red sentence: a sufficient and necessary condition;
• Green sentence: a sufficient but not necessary condition;
• Blue sentence: a necessary but not sufficient condition;
• Black sentence: a neither sufficient nor necessary condition.

Green box: GT, Red box: prediction by ILC, Pink box: predicted by ELC, Blue box: person detected by GroundingDINO but not selected by either ILC or ELC. pf: probability of prediction on full annotation by ILC, pm: probability of prediction on multiple sentences by ELC.

Full annotation 5:
The individual in question has been identified as a soldier, positioned in the middle of a group of three. He is dressed in military fatigues that feature the recognizable U.S. flag patch on his right arm, and his surname, 'Daquina,' is prominently displayed on his chest. The uniform he wears is the typical digital camouflage design, accessorized with a battle helmet, protective goggles set atop the helmet's edge, and a tactical vest. His hands are positioned behind his back.

• Red sentence: a sufficient and necessary condition;
• Green sentence: a sufficient but not necessary condition;
• Blue sentence: a necessary but not sufficient condition;
• Black sentence: a neither sufficient nor necessary condition.

Green box: GT, Red box: prediction by ILC, Pink box: predicted by ELC, Blue box: person detected by GroundingDINO but not selected by either ILC or ELC. pf: probability of prediction on full annotation by ILC, pm: probability of prediction on multiple sentences by ELC.

Figure 8: Examples from HC-RefLoCo.

score, the pink box and text indicate the prediction made by ELC, and the blue box and text indicate the detected referred object associated with ELC's prediction. Under each image, the phrase of referring expression is presented. At the bottom, the explanation of the boxes in the image are presented.

In the first example, ILC with EvaCLIP fails to distinct the tank top with white shirt and makes a wrong prediction. On the other hand, GroundingDINO distinguish the tank top correctly, leading to the right prediction by logic reasoning on the grounded object and spatial relation with the person. In the second example, it might be difficult to discriminate the 'black and gray shirt' with dark blue shirt as well as the relative length of hair. ILC on EvaCLIP makes a wrong prediction. However, GroundingDINO successfully grounds the 'black and gray shirt' and 'longer brown hair' in the image. Hence, ELC makes a right prediction on the explicit visual evidence. The last two examples come from the same image. It is found that both EvaCLIP and GroundingDINO struggle to discriminate 'blue color shirt' with 'baby blue shirt'. The decision on ILC's prediction is unexplainable. However, ELC gives explicit visual evidence on its decision, regardless of whether the answer is correct or wrong. These examples show that with explicit facts and relations, it is easy for us to make logic validation and justification on the final predictions on HC-REC task.

### 4.3    EXAMPLES FROM HC-REFLOCO

Four examples from HC-RefLoCo are presented in Figure 8, where ILC on EvaCLIP with full annotation and ELC on InternVL2(8b) with multiple sentences produce inconsistent predictions. In each image, the green box indicates the GT box, the red box and text indicates the prediction made by ILC with its probability, the pink box and text indicates the prediction made by ELC, and the blue box and text indicates other detected persons. On the right side of the image, the long context full annotation is presented. As indicated below the full annotation, on the logic cause-effect relations, the green sentence could be assumed as a sufficient but not necessary condition, the blue sentence could be assumed as a necessary but not sufficient condition, the red sentence could be considered

as a sufficient and necessary condition, and the black sentence can be assumed as a neither sufficient nor necessary condition. At the bottom, the explanations of the boxes in the image are presented.

In the two examples of the same image, it can be found that the green sentences would lead the attention to anyone of the three US soldiers. Hence, they could be considered as a sufficient cause to find the target person in the image, but not the necessary causes. The red sentences contain distinctive descriptions which would lead to the target person uniquely, and can be considered as a sufficient and necessary cause. The blue sentences contain the unique object(s) associated to the target person. Hence, it can be assumed as a necessary but not sufficient cause which can help to localize the target person. The black sentences are usually less related to the target person in the HC-REC task. Hence, they can be considered as neither necessary nor sufficient conditions.

In the first example, the ILC on EvaCLIP with full annotation as text input fails to localize the right person, as indicated by the red box. It may be confused by the green sentences and lose the cues for right prediction. However, the ELC on InternVL2(8b) makes prediction based on weighted sum of sentence categories, is able to correctly localize the target person, showing the effectiveness of logic reasoning on sentences for HC-REC task. In the second example, the first red sentence is not very effective in our experiment as the person bounding box is cropped and fed to LVLMs. The spatial cues may be lost. Hence, for the prediction based on split sentences, the sufficient and necessary conditions are weak. While on the full annotation, ILC on EvaCLIP makes a correct prediction, which maybe less affected by the truncation of the relatively shorter full annotation. The ELC on sentences fails to localize the correct person, due to the lack of sentences with both sufficient and necessary conditions, except for the first sentence with spatial cue information. However, it makes the second highest prediction on the correct person. On these two examples, the integrated prediction by LCR produces the right predictions, by fusing the strengths of both ILC and ELC.