# OpenReview forum: "Logic Channel Validation and Enhancement of Zero-Shot Vision-Language Comprehension on Vision Language Models"
_ICLR.cc/2026/Conference — Submitted to ICLR 2026_

### Official Review · Reviewer_yfrb · 2025-10-28

**Soundness:** 2
**Presentation:** 3
**Contribution:** 2
**Rating:** 4
**Confidence:** 2

**Summary:**

This paper presents Logic Channel Validation (LCV), a framework for evaluating and improving reasoning in large vision-language models by separating their decision process into two complementary parts. The Implicit Logic Channel (ILC) captures how a VLM internally reasons when directly judging an image–text pair, while the Explicit Logic Channel (ELC) explicitly parses the linguistic structure using a language model and grounds the visual entities and relations with a vision foundation model through probabilistic logic. The paper introduces Channel Reliability (CR) to measure the consistency between implicit and explicit reasoning and evaluates the approach on three benchmarks.

**Strengths:**

1. The logic decomposition feels natural and intuitive to use in practice.

2. A clear formulation with a simple CR metric enabling label-free validation and model selection.

3. A modular pipeline (LLM parsing + VFM grounding + probabilistic logic) that yields interpretable evidence and easy ablations.

**Weaknesses:**

1. The claimed **comprehensive** scope is narrow, since ELC mostly covers existence/negation/spatial relations and not attributes or OCR, so the paper should clarify boundaries and outline extensions.
2. The approach is heavily dependent on the chosen VFM, so a cross-VFM sensitivity check is needed to establish robustness.
3. ILC relies on self-reported 0–100% confidences without reporting decoding settings or calibration, so the paper should include temperature/prompt details.
4. CR can be inflated by correlated errors between channels, so the paper should decompose agree-and-correct vs agree-and-wrong and relate CR to human accuracy.

**Questions:**

1. Could you clarify the actual reasoning scope of ELC? Beyond existence, negation, and spatial relations, can it handle attributes or OCR cases, and if not, what are your plans to extend it?

2. How sensitive is ELC to the underlying vision foundation model? Have you tested robustness across different detectors?

3. What were the decoding and prompting settings when obtaining ILC’s 0–100% confidences (e.g., temperature), and how stable are the results under different settings or paraphrased prompts?

---

> ### Author Response · Authors · 2025-11-28
>
> Weaknesses:
>
> W1. Yes, visual-language comprehension (VLC) is a general capability of human intelligence. Existing approaches to benchmark the VLC abilities of LVLMs are by building benchmark datasets with various VLC tasks for training and evaluation. The basic VLC tasks in literature are VQA and Visual Grounding. The general logic inference skills discussed in literature are factual, counter-factual, relationship, etc. In this paper, we investigate both tasks of VQA and Visual Grounding, where NegBench is a recent VQA challenge with negations, and HC-Ref tasks focuses on human related activities. Yes, if additional VLC functions are required, such as OCR recognition for VQA on documents, the SOTA VFM on OCR (e.g., DeepSeekOCR) has to be introduced, and basic logic rules on OCR understanding have to be implemented. In this paper, general logic inference rules are used to show the effectiveness of our method, especially for zero-shot validation without gt annotation. In real-world applications, additional logic rules would be implemented on task requirement, such as applications to finance, law and industries.
>
> W2. The LVLM, VFM and LLM models may keep changing due to rapid progress of frontier LLMs and MLLMs. When applied for a new task, latest VFM can be employed. The representative VFMs for visual grounding are currently GroundingDINO series and SAM thread. We have tested both models. GroundingDINO directly predicts the bounding box, while SAM produces the pixel-level segmentation first and then derives the bounding box. Naturally, the task for SAM is more challenging and difficult than the task for GroundingDINO. As reported in the benchmarking results in HC-RefLoCo, the performance of cluster on segmentation (e.g., SAM) is lower than the cluster on detection. GroundingDINO is selected based on this investigation. Ablation study on comparison with SAM can be provided. In addition, ILC and ELC cross-validation on CR could be used for selection of VFM (please see reply to W2 of Reviewer RcoM).
>
> W3. Prompting methods are described in detail in the Supplementary Material. We adopt the default prompt format and setup provided by the corresponding benchmarks. For LVLMs (i.e., EvaCLIP and InternVL2), we employ the default setting provided with the models, e.g., for InternVL2-8B, we employ the default temperature (0.8).
>
> W4. CR is proposed for validation w/o gt annotation. With gt annotation, we can obtain CRwgt and statistics of win-win (agree-and-correct), win-lose, lose-win, and lose-lose, as well as lose-lose with the same choice (i.e., agree-and-wrong). Please see the tables below, and tables in reply to Reviewer 1ZfW. In the tables below, Rww means Rate of win-win (both ILC and ELC select the correct answer on gt), Rwl is for Rate of win-lose (ILC chooses the correct answer, but ELC chooses a wrong answer), Rlw is for Rate of lose-win (ILC chooses a wrong answer but ELC chooses the correct answer), Rlls is for Rate of lose-lose-same (both ILC and ELC select a same answer but it is a wrong answer), and Rlld is for Rate of lose-lose-dif (ILC and ELC chooses two different answers but both are wrong answers).  Yes, CR is also helpful to check human accuracy. In the Supplementary Material, we provide examples where ILC selects the choice of gt annotated by human, while ELC selects another choice, but the explicitly grounded visual facts show that the ELC’s choice is the right answer. Human annotators might ignore the related visual cues and ILC makes decision on hallucination.
>
> |**NegBench**|||**COCO**||||
> |:---|:---:|:---:|:---:|:---:|:---:|:---:|
> |Model|CR|Rww|Rwl|Rlw|Rlls|Rlld|
> |EvaCLIP-Logic|27.05|19.61|5.21|56.15|7.44|11.58|
> |InternVL2-Logic|62.80|53.47|9.79|22.30|9.33|5.11|
> ||||**VOC2007**||||
> |EvaCLIP-Logic|27.25|23.47|3.12|63.37|3.78|6.26|
> |InternVL2-Logic|79.25|76.29|9.12|10.55|2.96|1.07|
> ||||||||
>
>
> |**Benchmark**|||**HC-RefCOCO**||||
> |:---|:---:|:---:|:---:|:---:|:---:|:---:|
> |Model|CR|Rww|Rwl|Rlw|Rlls|Rlld|
> |EvaCLIP-Logic|35.09|26.32|21.34|20.41|8.77|23.18|
> |InternVL2-Logic|23.04|18.59|14.22|28.14|4.46|34.59|
> ||||**HC-RefCOCO+**||||
> |EvaCLIP-Logic|40.37|33.33|22.09|20.20|7.04|17.33|
> |InternVL2-Logic|31.77|27.23|16.36|26.17|4.40|25.70|
> ||||**HC-RefCOCOg**||||
> |EvaCLIP-Logic|63.12|56.19|21.28|10.47|6.93|5.12|
> |InternVL2-Logic|40.77|36.82|11.88|29.84|3.94|17.51|
> ||||||||
>
>
> |**HC-RefLoCo**|||**Val**||||
> |:---|:---:|:---:|:---:|:---:|:---:|:---:|
> |Model|CR|Rww|Rwl|Rlw|Rlls|Rlld|
> |ILC_EvaCLIP-ELC_EvaCLIP|72.90|59.28|15.01|7.18|13.62|4.91|
> |ILC_InternVL2-ELC_InternVL2|18.48|13.19|2.70|63.47|5.29|15.35|
> |**HC-RefLoCo**|||**Test**||||
> |ILC_EvaCLIP-ELC_EvaCLIP|73.91|59.40|15.19|6.42|14.50|4.49|
> |ILC_InternVL2-ELC_InternVL2|18.48|13.19|1.93|66.15|2.18|16.16|
> ||||||||

---

> > ### Author Response · Authors · 2025-11-28
> >
> > Questions:
> >
> > Q1. Please see the reply to W1 above.
> >
> > Q2. Please see the reply to W2 above.
> >
> > Q3. Prompting methods are provided in the Supplementary Material. We adapt the prompt protocol provided by the corresponding benchmarks. Yes, if the prompt format is quite different from the default format provided by the builder of benchmark, one may obtain quite different results. The effects of prompt variations to LLM and MLLM have been well-studied. We can cite a few representative surveys on it in the revised paper. As we target zero-shot validation w/o gt annotation, no additional decoding is added.

---

### Official Review · Reviewer_RcoM · 2025-10-28

**Soundness:** 2
**Presentation:** 2
**Contribution:** 2
**Rating:** 4
**Confidence:** 3

**Summary:**

This paper introduces the Logic Channel framework to address the lack of trustworthiness and explainability in frontier LVLMs, especially for zero-shot tasks. The core idea is a dual-channel system: an implicit channel (the black-box LVLM) runs parallel to an Explicit Logic Channel (ELC). The ELC mimics human reasoning by using an LLM to parse sentences, a VFM to ground key facts, and a set of customized probabilistic logic rules for inference. The Logic Channel serves two purposes: validation, assessing model reliability via cross-channel consistency checks (CR), and enhancement, boosting SOTA accuracy on logical reasoning benchmarks like HC-RefCOCOg by fusing the two channel predictions.

**Strengths:**

1. The paper identifies the critical gap where current LVLMs are black boxes lacking reliability. By proposing an explicit, explainable logic channel to serve as a verifiable "auditor" for the black-box LVLM, it offers a direct and timely solution to enhance model trust and transparency.

2. By combining predictions from both the ILC and ELC, the system reliably improves the final decision accuracy. This enhancement is notable on complex logical reasoning benchmarks (like HC-RefCOCOg), validating the benefit of explicit logic in unlocking LVLM potential.

**Weaknesses:**

1. The paper designs separate, customized probabilistic inference formulas for each specific logic type (negation, relation, causation). This strongly suggests the ELC's logic module lacks true generalizability. If a new, mixed-logic task emerges, researchers may have to manually write new logic formulas, severely limiting the framework's automation and scalability.

2. The ELC is a complex pipeline assembled from pre-trained black-box components (LLM parser, VFM detector, logic module). Its robustness is constrained by the weakest link. If VFM fact grounding fails (e.g., missing a crucial object), the entire explicit reasoning chain breaks. The paper lacks a proper robustness analysis against these cascading component failures.

3. The ELC operates as a parallel system requiring sequential calls to an LLM and a VFM, introducing a significant computational and latency overhead. The paper completely fails to quantify this cost, which makes the "enhancement" feature's practicality for real-time deployment highly questionable.

**Questions:**

Q1. The paper shows success across limited logic forms (based on existence/relation/causal categories). If a mixed-logic and complex task requiring negation, relation, and causation simultaneously were introduced, how would the ELC's logic module automatically or uniformly combine these separate probabilistic logics for a single final decision?

Q2. To assess the true deployment cost of the enhancement feature, please provide detailed inference latency data  for the ELC channel across the benchmarks. This breakdown should include the segmented time consumption for the LLM parsing, VFM grounding, and final logic reasoning.

---

> ### Author Response · Authors · 2025-11-28
>
> Weaknesses:
>
> W1. The purpose of this paper is not to propose another general VL method to compete or replace frontier LVLM models, but rather to provide a framework with a logic channel for explicitly validating and justifying the decision-making by LVLM, especially for zero-shot application in a new task without ground-truth annotation. The LVLM, VFM and LLM models may keep changing due to rapid progresses of frontier LLMs and MLLMs. When applied for a new task, under our proposed framework, latest LLM, VFM and LVLM can be employed, and new logic rules beyond the general logic inference implemented here can be implemented in straightforwardly on visual facts and relations provided by VFM. It might be affordable for user compared to scaling-up the LVLM (e.g., frontier MLLM) with new collected and annotated dataset. In this paper, we show the implementation on basic logic rules of inference, i.e., factual, counter-factual, relationship, and strength of sufficiency and necessity of causality, which cover a wide range of basic logic inference. Our implementation for general logic reasoning on factual, counter-factual, and relationship have shown the generalization capabilities and scalability, and ease for adaption to additional logic rules for reasoning, especially when dataset and gt annotation are not available for retraining and systematic evaluation.
>
> W2. Yes, when deployed to new tasks, the poor performance of final prediction might be caused by weak capabilities of LVLM, VFM and LLM. However, the parallel channel framework would benefit cross-model validation between VFM and LVLM even without gt annotation. There may be four cases. (a) If LVLM and VFM both perform well on the task, there will be high logic consistency. There are few cases where ILC and ELC don’t agree with each other. The visual observation on explicit visual evidence would also support the conclusion; (b) If VFM performs well but LVLM performs poorly on the task, there would be increased cases of logic inconsistency, and manual examination on the cases will reveal the errors of hallucinations and missing visual cues. (c) if LVLM performs well but VFM performs poorly, there would be increased cases of logic inconsistency, and manual examination on the cases will reveal the errors of object detections by VFM or errors of language parsing by LLM. (d) if both LVLM and VFM perform poorly, naturally, the cases of logically inconsistency are high indicated by very low CR score, and manual examination will find such errors explicitly on hallucinations and object detection errors, which means SOTA LVLMs and VFMs are still not mature for the application domain even the formal evaluation on collected and gt annotation have not performed yet. Hence, our framework and method provide a practical way for explicit validation to understand the performance of LVLM and VFM on a new task even though a well annotated dataset is not ready.
>
> W3. As we target real-world applications, efficiency is very important. Hence, we focus on small MLLMs of 7B-8B size. The computational costs are: Mistral (LLM) for language parsing is 0.76sec (on HC-RefCOCOg with average 8.9 words), GroundingDINO (VFM) for human and object detection is 0.35sec, InternVL2 (MLLM) for ILC requires 0.472sec for each image to choose one from 4 choices for NegBench Datatset, and EvaCLIP requires 1.40sec on one VQA test in average over three benchmarks (NegBench, HC-RefCOCOg, and HC-RefLoCo). Logic reasoning requires almost no additional time cost compared with VFM and LVLM.
>
> Questions:
>
> Q1. When applied to a real-world task, first, on the input text query, the LLM is prompted to predict task requirement and required visual cues (nouns), the texts of nouns are fed to VFM to ground the objects in the image,  then, the ELC calls relevant logic reasoning functions and justifies the answer explicitly on visual evidence produced by VFM. For user to use it for a new task, the effective prompting format might to be designed when deploying our method for a real-world application task.
>
> Q2. Considering the efficiency of deployment in real-world applications, we evaluated on small MLLMs, i.e., 7B to 8B models. The average time of GroundingDINO (VFM) detection is 0.21sec for HC-RefLoCo (person only), 0.35 sec for HC-RefCOCOg (person + object), 0.76sec for NegBench (objects from 4 choices). Mistral (LLM) for language parsing is 0.76sec (on HC-RefCOCOg with average 8.9 words). For MLLM, e.g., InternVL2.0 is 0.472sec for each image to choose one from 4 choices for NegBench Datatset, EvaCLIP is 1.40sec on one VQA test in average over three benchmarks (NegBench, HC-RefCOCOg, and HC-RefLoCo).

---

### Official Review · Reviewer_xErW · 2025-10-30

**Soundness:** 3
**Presentation:** 3
**Contribution:** 1
**Rating:** 2
**Confidence:** 4

**Summary:**

To validate and improve the performance of visual language models across various visual comprehension tasks, the authors propose a framework that adds explicit logic channels (ELC) to the main model pipelines (referred to as implicit logic channels, i.e., the visual models under evaluation). In this setup, the ELC is a task-specific, carefully designed combination of LLMs and other foundation models (such as object detectors) that can perform the same task. The ELC’s outputs are then used either to validate the main model’s responses or to enhance them when errors occur.
In the methodology section, the authors describe their ELC designs for three tasks: generating correct text descriptions from images (logical reasoning on facts and negation), referring expression comprehension (logical reasoning on facts and relations), and referring expression comprehension with long context (logical reasoning on causal conditions). The evaluation explores how the ELC contributes to both validation and performance improvement across these tasks, showing that the inclusion of an ELC helps with model diagnosis, validation, selection, and overall output quality.

**Strengths:**

The paper addresses a relevant and timely problem and is written clearly. The assumptions are well laid out, and the experimental results generally support the authors’ claims. The overall presentation is easy to follow, and the technical details are described with care.

**Weaknesses:**

I do have some reservations regarding the level of contribution and the generalizability of the proposed framework. In essence, the authors design a set of expert-crafted pipelines (the ELCs) for the three target tasks and then use these as oracles to validate or improve the main visual-language model (VLLM). However, each ELC is itself a more complex, task-specific system that can already perform the task effectively. This raises the question of whether the ELC truly provides a meaningful or general solution—especially since the framework implicitly assumes that the ELC performs near-perfectly, in which case it might render the use of the main VLLM unnecessary.

Overall, while the paper presents a creative and well-structured framework, it does not fully convince me of the necessity or broader impact of the proposed approach. The idea is interesting and the results are promising, but the paper would benefit from a clearer motivation and stronger justification for why this framework is needed in the first place.

**Questions:**

N/A

---

> ### Author Response · Authors · 2025-11-26
>
> Weaknesses:
> The purpose of this paper is not to propose another AGI method to compete or replace existing LVLM models, but rather to provide a general framework to develop a practical approach for explicitly validating and justifying the decision-making by LVLM, especially for zero-shot application in a new task without ground-truth annotation. The ELC is built on general logic rules and evidence from SOTA VFM and LLM. In real-world applications, such as multimodal interactions in VQA in education and real-time industrial tasks, the re-training and fine-tuning of LVLM in advance of deployment would be prohibitive due to data privacy, and systematic dataset collection and annotation is not applicable due to manpower constraints. The LVLM, VFM and LLM models may keep changing due to rapid progresses of frontier LLMs and MLLMs, our method provides a continual way to develop an ELC for validation and justification of LVLM’s prediction with explicit visual evidence and support. We have prepared Supplementary Material with examples to show the effectiveness to address (a) hallucination, (b) ambiguity, (c) and weak power of VFM for a new task. For real-world new tasks, additional VFM model might be involved, such as VFM for OCR for VQA on industrial manuals, and new logic rules might be implemented, however, such efforts are affordable for user compared to scale-up the LVLM (e.g., frontier MLLM) with new collected and annotated dataset. This method is motivated by real-world applications. We evaluated it on public benchmarks to show the effectiveness on concrete evaluations as well as for publication purposes. We have expressed the motivations of this work in Abstract and Induction, and will further enhance it on the reviewer’s suggestion.

---

### Official Review · Reviewer_1ZfW · 2025-11-01

**Soundness:** 3
**Presentation:** 3
**Contribution:** 2
**Rating:** 6
**Confidence:** 4

**Summary:**

This paper proposes a training-free method for Visual-Language Comprehension (VLC) tasks that leverages an LLM to extract entities/relations from textual descriptions, a Vision model as a detector for grounding (text-to-image), and a logical reasoning module for probabilistic inference of factual, counterfactual, relational, and causal condition reasoning over the extracted and grounded image-text. The authors evaluate their proposed method on NegBench MCQ and HC-RefCOCOg / HC-RefLoCo referring tasks.

**Strengths:**

- The proposed pipeline effectively decompose the image and text inputs, which enable further explainability and interpretability. The setup is simple, yet effective, and is consistent with related work that also tackles visual reasoning through similar decomposition pipelines.

- This paper covers negations, relations and long context text, and show strong results of using off-the-shelf models for textual decomposition and visual grounding without finetuning for visual-language comprehension.

- Strong results on NegBench (negation) and HC-RefLoCo (long-context). This expands prior works results on compositional benchmarks to other challenging tasks (particularly the long-context setup)

**Weaknesses:**

- Missing prior works (novelty and claims): The paper’s idea of LLM extraction/expansion, symbolic/probabilistic solver and detector grounding is very close to LINC [1], BIRD [2], and other neuro-symbolic pipelines that also use visual entailment + contradictions (factual + counterfactual, neutral + negations) for image-text evaluation/alignment [3, 4, 5, 6]. Particularly, [4, 7] show strong performance with training-free strategies.

- Missing ablations for the parser LLM (prompting, temperature), the VFM (GroundingDINO thresholds, NMS, text queries), sentence-utility weights for the long-context setting, and fusion weights across channels. There are no ablations for swapping VFMs (e.g., OWL-ViT, Florence-2), parser quality/noise injection, probability aggregation rules (min/max vs. soft aggregators), or final calibration.

-

[1] LINC: A Neurosymbolic Approach for Logical Reasoning by Combining Language Models with First-Order Logic Provers (Olausson et al., EMNLP 2023)

[2] Feng, Yu, et al. "Bird: A trustworthy bayesian inference framework for large language models." arXiv preprint arXiv:2404.12494 (2024).

[3] Yarom, Michal, et al. "What you see is what you read? improving text-image alignment evaluation." Advances in Neural Information Processing Systems 36 (2023): 1601-1619.

[4] CounterCurate: Enhancing Physical and Semantic Visio-Linguistic Compositional Reasoning via Counterfactual Examples (Zhang et al., ACL Findings 2024)

[5] Wang, Tan, et al. "Equivariant similarity for vision-language foundation models." Proceedings of the IEEE/CVF International Conference on Computer Vision. 2023.

[6] Kamath, Amita, et al. "The hard positive truth about vision-language compositionality." European Conference on Computer Vision. Cham: Springer Nature Switzerland, 2024.

[7] Cascante-Bonilla, Paola, et al. "Natural language inference improves compositionality in vision-language models." ICLR 2025.

**Questions:**

- High agreement between two correlated channels can be confidently wrong; is there any empirical/quantitative analysis about the correlation between the Consistency Rate and accuracy across models/datasets?

---

> ### Author Response · Authors · 2025-11-29
>
> Weaknesses:
>
> W1. Thanks for the list of additional prior works. Some papers are discussed in Related Work. LINC [1] and BIRD [2] has been discussed in the cluster of “Logic reasoning on LLM and VLM” from line 105 to 119. In [3] and [4], new datasets SeeTRUE and CounterCurate with both positive and negative questions are proposed, similar to NegBench experimented in this paper, while in NegBench a sample has one positive and three negative queries on each image. The setting of a pair of images and questions with alternating answers is similar to NaturalBench (NeurIPS24). However, the yes/no setting has 50% chance of random selection. In [5], a regularization loss EQSIM is introduced for training to improve VLM to the challenge of equivariant similarity. In [6], hard positive and hard negative captions are introduced by word swapping and replacement on original caption for training, which leads to improved robustness of the VLMs. In [7], a CECE approach is proposed which extends an image-question pair with additional entailment set and contradiction set, and computes the balanced assessment on the all image-question pairs to enhance the robustness, similar to the self-consistency on a cluster of neighbouring questions in one paper in CVPR2025. Our paper focuses on zero-shot validation of LVLM on explicit visual evidence and logic reasoning without gt annotation.
>
> W2: Our experiments focus on showing the effectiveness of our method on existing models for zero-shot validation w/o gt annotation. Benchmarking the accuracy performance is not the main purpose of this paper. Prompting methods are described in detail in the Supplementary Material. We adopt the default prompt format and setup provided by the corresponding benchmarks. For LVLMs (i.e., EvaCLIP and InternVL2), we employ the default setting provided with the models, e.g., for InternVL2-8B, we employ the default temperature (0.8). Many experimental details are prepared in the Supplementary Material, such as sentence-utility weights for long-context questions. In main paper, we have mentioned such information, but we missed the submission of Supplementary Material. Thanks for the suggestions, the details of suggested ablation studies to be presented in Supplementary Material.
>
> Questions:
>
> Q1: The CR (Consistency Rate) metric is defined without gt annotation, aiming for the application of zero-shot evaluation without gt annotation. With gt annotation, we can obtain the Consistency Rate of correct answers (CR_wgt). The comparison between CR and CR_wgt on the three benchmarks are reported in the tables below. In general, most (at least over 70%) consistent answers are correct answers, and, the higher the CR, the higher the rate of correct answers of consistent answers (e.g., CR=79.25%, the rate of CR_wgt answers in CR answers is 96.26%).
>
> |**NegBench**||**COCO**|||**VOC2007**||
> |:---|:---:|:---:|:---:|:---:|:---:|:---:|
> |Model|CR|CR_wgt|Rate|CR|CR_wgt|Rate|
> |EvaCLIP-Logic|27.05|19.61|**72.50**|27.25|23.47|**86.13**|
> |InternVL2-Logic|62.80|53.47|**85.14**|79.25|76.29|**96.26**|
> |Qwen2.5-Logic|70.73|61.85|**87.45**|83.50|81.42|**97.51**|
> ||||||||
>
> |||**HC-RefCOCO**|||**HC-RefCOCO+**|||**HC-RefCOCOg**||
> |:---|:---:|:---:|:---:|:---:|:---:|:---:|:---:|:---:|:---:|
> |Model|CR|CR_wgt|Rate|CR|CR_wgt|Rate|CR|CR_wgt|Rate|
> |EvaCLIP-Logic|35.09|26.32|**75.09**|40.37|33.33|**82.56**|63.12|56.19|**89.02**|
> |InternVL2-Logic|23.04|18.59|**80.69**|31.77|27.23|**85.71**|40.77|36.82|**90.31**|
> |||||||||||
>
> |**HC-RefLoCo**||**Val**|||**Test**||
> |:---|:---:|:---:|:---:|:---:|:---:|:---:|
> |Model|CR|CR_wgt|Rate|CR|CR_wgt|Rate|
> |ILC_EvaCLIP-ELC_EvaCLIP|72.90|59.28|**81.32**|73.91|59.40|**80.37**|
> |ILC_InternVL2-ELC_InternVL2|18.48|13.19|**71.37**|15.76|13.58|**86.17**|
> ||||||||

---

### Meta-Review · Area_Chair_9Wi6 · 2026-01-06

**Summary:**

The paper proposes a "Logic Channel" framework to validate and enhance Vision-Language Models (VLMs) in zero-shot settings. The core approach involves a dual-channel system: an "Implicit Logic Channel" (the VLM itself) and an "Explicit Logic Channel" (ELC) which pipelines an LLM parser, a Vision Foundation Model (VFM), and a probabilistic logic module. While the reviewers appreciated the motivation to improve interpretability and the solid performance on specific benchmarks like NegBench, the consensus leans towards rejection due to several fundamental limitations. First, the novelty is questioned, as the ELC essentially constructs a modular neuro-symbolic pipeline (combining LLMs and VFMs) similar to existing works like LINC or BIRD, which reviewers felt was not adequately distinguished. Second, the scalability and generalizability of the "logic" component are limited; reviewers noted that the reliance on hand-crafted probabilistic formulas for specific logic types (negation, relation, causal) hinders the method's ability to handle mixed or novel logic tasks without manual intervention. Third, the robustness of the ELC is constrained by the cascading errors of its components (e.g., VFM grounding failures), making the validation channel potentially as brittle as the model it attempts to validate. Although the authors provided a detailed rebuttal addressing latency and defining the scope of zero-shot validation, the concern regarding the "engineering heavy" nature of the solution—where the validation mechanism (ELC) is arguably just a separate, complex model pipeline—remains a barrier to acceptance at ICLR.

**Reviewer Concerns:**

Addressed Concerns: (1) Latency: The authors provided specific inference times for the ELC components (e.g., Mistral + GroundingDINO) in response to Reviewer RcoM, clarifying the cost of the explicit channel. (2) Zero-Shot Scope: The authors clarified that their primary goal is validation in scenarios where ground truth is unavailable (privacy/data constraints), helping to contextualize the method against supervised baselines (Response to Reviewer 1ZfW and xErW).

Outstanding Concerns: (1) Scalability of Logic Rules: Reviewer RcoM's concern about the manual design of probabilistic formulas for different logic types remains a core weakness. The rebuttal confirmed that new rules must be implemented for new tasks, confirming the lack of automation/generalization. (2) Fundamental Necessity: Reviewer xErW’s critique that the ELC is effectively a "better" task-specific pipeline that renders the main VLM redundant in some cases was not fully resolved. The rebuttal argued for "validation," but if the ELC is the "gold standard," it raises the question of why the VLM is needed for these specific reasoning tasks at all. (3) Cascading Failures: The reliance on the VFM (GroundingDINO/SAM) means the "logic" validation is entirely dependent on the detector's quality. If the VFM fails to ground an object, the logic channel fails. This structural brittleness (noted by RcoM and yfrb) is inherent to the pipeline design and was not structurally addressed.

**Reviewer Scores:**

Reviewer 1ZfW: 6 --> 5

Reviewer xErW: 2 --> 3

Reviewer RcoM: 4 --> 4

Reviewer yfrb: 4 --> 4

---

### Decision · Program_Chairs · 2026-01-26

Reject